# Biomass Estimation and Carbon Storage of *Taxodium* Hybrid Zhongshanshan Plantations in the Yangtze River Basin

Qin Shi [1], Jianfeng Hua [1], David Creech [2] and Yunlong Yin [1,*]

1   Jiangsu Key Laboratory for the Research and Utilization of Plant Resources, Institute of Botany, Jiangsu Province and Chinese Academy of Sciences (Nanjing Botanical Garden Mem. Sun Yat-Sen), No. 1 Qianhuhou Village, Nanjing 210014, China
2   Arthur Temple College of Forestry and Agriculture, Stephen F. Austin State University, No. 1936 North Street, Nacogdoches, TX 75962-3000, USA
*   Correspondence: yinyl066@sina.com; Tel.: +86-025-8434-7143

**Abstract:** As a pivotal wetland tree, *Taxodium* hybrid Zhongshanshan has been widely planted in the region of Yangtze River for multipurpose of ecological restoration, field shelter, landscape aesthetics as well as carbon sequestration. However, the carbon allocation patterns across distinct stages of stand development of *T.* Zhongshanshan are poorly documented. Using a sample of 30 trees which were destructively harvested, this study compared 3 models for assessing aboveground biomass. Furthermore, a linear seemingly unrelated regression (SUR) approach was introduced to fit the system of the best selected model that ensured the additivity property. On this basis, biomass and carbon storage of *T.* Zhongshanshan stands in the Yangtze River Basin (YRB) were fairly estimated. Specifically, the study developed height-diameter at breast (*H-DBH*) function. The results showed that the selected 3-parameter polynomial model performed better, and the SUR approach provided more accurate estimates of leaf and stem fractions. The total tree biomass was 53.43, 84.87, 140.67, 192.71 and 156.65 t ha$^{-1}$ in the 9-, 11-, 13-, 15-, and 22-year-old *T.* Zhongshanshan stands, and contributed averagely 94.40% of the ecosystem biomass accumulation. The current *T.* Zhongshanshan stands in the YRB area can store 124.76 to 217.64 t ha$^{-1}$ carbon, of which total tree ranges from 25.32 to 90.89 t ha$^{-1}$, with 55.19% to 77.66% storing in the soil. The *T.* Zhongshanshan had continuous potential for carbon storage during its growth, particularly in the incipient stages. The findings of this research are firsthand information for forest managers for the sustainable management of *T.* Zhongshanshan in the YRB and similar subtropical areas.

**Keywords:** biomass estimation; carbon storage; allometric model; *Taxodium* hybrid Zhongshanshan; Yangtze River Basin

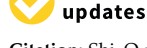



## 1. Introduction

Increasing global carbon fixation through the expansion of afforestation lands has been proposed as an efficient approach for alleviating elevated concentrations of atmospheric $CO_2$ [1]. The absolute and relative distribution of carbon storage in plantations is acquainted with a variation of tree species, soil condition and climate. Immediate challenges of forest management are concerned with credible, precise and cost-effective methods to adequately document forest dynamics. As a result of the monetary value being attached to carbon sequestration, there is increased scrutiny of techniques for estimating tree biomass. Burgeoning carbon credit market mechanism such as reducing emission from deforestation and forest degradation necessitates such a need [2]. This requires appropriate methods specific for a given forest type. Traditionally, allometric relationships to aboveground biomass have been used at the experimental scale to estimate aboveground biomass. Generic equations, stratified by ecological zones, for estimating aboveground biomass exist, but they may not accurately reflect the tree biomass in a specific area or

region [3]. Small tree individuals not merchantable are often omitted in forest resource investigation.

The allometric model is a statistical formula calculated by regression analysis between tree properties, among which tree diameter at breast height (*DBH*) and height (*H*) have often been used as explanatory variables because of their immediate availability. Developing such relationships of a certain tree species is a time-consuming activity, especially separating biomass components including the leaves and branches. A good database for developing regression equations should contain an age sequence because trees of different diameters distinguish from each other in the component proportion of the aboveground biomass. When there were several tree components in the biomass data, the additivity of models for accessing tree total, sub-total, and separate biomass fractions should be considered, owing to the inherent correlations among the biomass components measured on the same sample trees [4]. Traditional models often ignored such inherent relationships. The seemingly unrelated regression (SUR) model, proposed by Parresol et al. [5], has been adopted by many reports to facilitate biomass models construction since it ensures the additivity among components and total biomass predictions [6,7]. Conventionally, forest inventories measure the *DBH* of all trees in each plot but often few are randomly selected and measured for *H*. Accurate measurement of tree *H* is more difficult than *DBH* measurement [8], which implies that the forest productivity appraisal, in practice, requires *H-DBH* models for *H* estimation. Moreover, in the cases where the actual measurements of height growth are not available, *H-DBH* functions can be used to indirectly predict height growth.

*Taxodium* hybrid Zhongshanshan (*T. distichum* × *T. mucronatum*), a superior inter-species hybrid, was successfully planted in southeastern China. Being one of the most important tree species, *T.* Zhongshanshan produces excellent quality timber, with a tower-shaped morphological structure, high resistance of bending and cracking, and impressive water tolerance traits. These phenotypic characteristics enhance wind resistance and permit better performance in hostile coastal environments. Owing to its high commercial and ecological value, the artificially established area of *T.* Zhongshanshan in the Yangtze River Basin (YRB) is around 70,000 ha, with quantities over 50 million, and the demand for this conifer tree in landscape plantations and ecological restoration surpasses supply. Despite the fact that an increasing number of studies have been conducted from many perspectives, including crossbreeding, water resistance mechanism and photosynthetic traits [9–12], little information is available on carbon pools with stand ages in *T.* hybrid Zhongshanshan plantations since models to assist management of this species are in most cases lacking. An assessment of carbon storage in *T.* Zhongshanshan plantations is crucial for regional-scale evaluation of carbon dynamics and ameliorating these estimates requires plentiful field studies. The aims of this study were (1) to establish the allometric biomass equations for *T.* Zhongshanshan and its individual component biomass with consideration of stand age; and (2) to estimate the carbon storage of the *T.* Zhongshanshan ecosystem.

## 2. Materials and Methods

### 2.1. Study Sites Description

The forests both natural and artificial are rigorously conserved in the YRB, and *T.* Zhongshanshan plants should be sampled without felling or seriously damaging them. Given this, the destructive tree sampling for allometric models was conducted at the experimental base of Institute of Botany, Jiangsu Province and Chinese Academy of Sciences, which is located in the Qixia and Liuhe Districts, Nanjing and Tinghu District, Yancheng (Figure 1). These areas experience a typical subtropical monsoon climate. The mean annual temperature is 15.8 °C and the mean annual precipitation is 1085 mm, of which 65% falls during June-September. The study areas of biomass and carbon storage are located along the Yangtze River, respectively, in the cities of Nanjing, Jingzhou, Chongqing and Kunming (Figure 1). The elevation ranges from 5 m to 1823 m above sea level. Characterized by a subtropical monsoon climate, the annual average temperature ranges from 12.6 to 18.0 °C and mean annual precipitation is about 1076 mm. The areas of the *T.* Zhongshanshan stands

cover from 42 to 201 ha and are managed by the local forestry departments. Understory vegetation was dominated by common herbaceous species included: *Erigeron annuus*, *Alternanthera philoxeroides*, *Achyranthes bidentata*, *Chenopodium album*, *Rubus hirsutus*, *Angelica sieboldin*, *Mazus japonicus*, *Plantago depressa*, *Setaria viridis* and *Solidago canadensis*, with sparsely scattered woods (mainly *Morus alba* and *Ulmus pumila*).

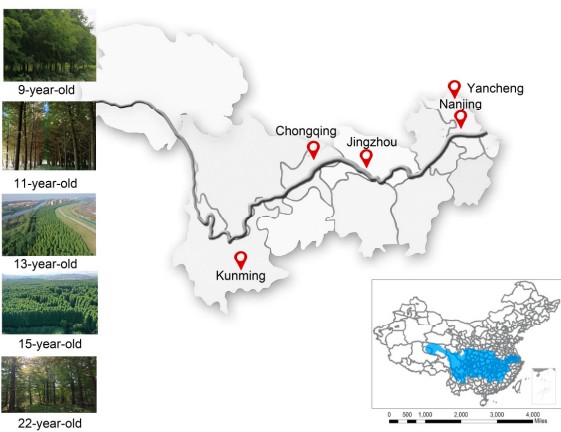

**Figure 1.** Location of the *T*. Zhongshanshan stands.

## 2.2. Destructive Tree Sampling

From April 2021 to June 2022, 30 trees aged 8, 12, 18 and 22 years (6–8 trees per age group) in the experimental base were used to establish the allometric equation for *T*. Zhongshanshan using the segmenting method. The tree was cut by an electric saw from the bottom. After *H* and *DBH* measurement, they were cut at 1 m intervals from the tree base. Each section was separated into stem, branch and leaf. The fresh weights of all components were measured in situ, and samples of every component in each standard tree were collected for water content and carbon concentration analysis. Allometric regression equations relating tree *DBH* and *H* were developed across this chronosequence, which was used to calculate the biomass of each tree and the total biomass of each *T*. Zhongshanshan stand.

## 2.3. Forest Inventory and Measurements

From May to June 2022, preliminary forest inventories were carried out in 9-, 11-, 13-, 15- and 22-year-old *T*. Zhongshanshan stands. We established a size of 1 ha covering minor local heterogeneity in soil and plant conditions, which allowed us to differentiate 4 fixed plots of 25 × 25 m in each stand and were spaced 30 m from each other. After measuring the distance of 25 m in a straight line with a 100-m tape, we inserted horticultural tallies into the 4 corners of each plot and connected them with strings to mark the boundaries. At each sampling stand, coordinate was recorded using a GPS device (A8, Zuolin Technology Co., LTD, Guangdong, China). Within each plot, *H* and *DBH* were recorded by a height measuring device (CGQ-1, Harbin Optical Instrument Factory, Harbin, China) and a professional tapeline for every tree (Table 1). The measured trees were marked with paint to guarantee that no repetition was made. All species were identified by 2 observers working together by randomly selecting five 1 m × 1 m quadrants in each plot. Litter, herb and shrubs biomass, including roots, were also harvested from these 5 subplots. All samples including tree tissues (leaves, branches and stems), herbs, shrubs and litter were weighed and oven-dried at 75 °C to a constant weight in the lab and reweighed through wet-to-dry mass conversion factors.

In each plot, 5 soil cores (5 cm in diameter) were randomly collected at 0–20, 20–40, and 40–60 cm depth. In view of the homogeneous soil within the stand, soil samples within the same layer in one plot were thoroughly mixed into a homogenized sample. Another set of soil samples from 0–20, 20–40, and 40–60 cm depths were separately and intactly collected

by inserting a steel cylinder of known volume (5 cm high and with a 5 cm inner diameter) for bulk density determination (ratio of dry mass to sample volume). All soil samples were put into labelled airtight plastic bags and taken back to the lab. A total of 60 soil samples and 60 bulk density samples were collected (5 stand ages × 4 plots × 3 depths). Moreover, a part of field samples was oven-dried at 105 °C for 24 h to determine soil moisture content gravimetrically. After removing the plant roots, fauna, and debris by hand, the soils were air-dried at room temperature, and then ground and passed through a 2-mm sieve for measurement of carbon contents. The carbon concentrations of the components from the tree, ground vegetation, forest floor and soil organic carbon (SOC) were determined by the potassium dichromate oxidation method.

**Table 1.** Characteristics of the 9-, 11-, 13-,15- and 22-year-old *T.* Zhongshanshan stands.

| Age (a) | Site | Location | Altitude (m) | *H* (m) | *DBH* (cm) | Density (Stems ha$^{-1}$) |
|---|---|---|---|---|---|---|
| 9 | Nanjing | 118°49′35″ E 32°10′59″ N | 15 | 8.4 | 13.6 | 1111 |
| 11 | Jingzhou | 112°13′37″ E 30°19′34″ N | 34 | 9.6 | 14.1 | 1111 |
| 13 | Chonqing | 108°27′3″ E 30°45′58″ N | 245 | 11.6 | 17.6 | 920 |
| 15 | Yunnan | 102°46′41″ E 24°49′43″ N | 1891 | 12.9 | 23.2 | 830 |
| 22 | Nanjing | 118°47′40″ E 32°20′13″ N | 17 | 14.6 | 33.1 | 410 |

*H*: height; *DBH*: diameter at breast height.

*2.4. Height-Diameter at Breast Height Function Development*

A nonlinear function below was used to model *H* for the sample tree measured for both *H* and *DBH* [13,14]. This function had the flexibility to produce satisfactory curves under most circumstances.

$$H = 1.3 + a \times [\exp(-b/DBH)]$$

where *a* and *b* are the parameters to be estimated.

*2.5. Tree Biomass Model Development*

In this study, the direct prediction of the tree biomass from measurement variables (*DBH* and *H*) was used to estimate the tree biomass of *T.* Zhongshanshan. Crown diameter is often quite irregular, even for a given species, depending on ecological conditions, and was not exploited further at this stage. We modeled tree biomass (*M*, in kg) as a function of *H* (in meters) and *DBH* (in cm) with the following 3 age-independent equations:

$$M = a \times (DBH)^b \tag{1}$$

$$M = a \times (DBH)^b \times H^c \tag{2}$$

$$M = a \times [(DBH)^2 \times H]^b \tag{3}$$

It is expedient to take logarithms for fitting the models and dealing with heterocedasticity. Therefore, Equations (1)–(3) can be linearized using logarithms in the following equations:

$$\ln(M) = a + b \times \ln(DBH) \tag{4}$$

$$\ln(M) = a + b \times \ln(DBH) + c \times \ln(H) \tag{5}$$

$$\ln(M) = a + b \times \ln[(DBH)^2 \times H] \tag{6}$$

where *M* is tree biomass, and *a*, *b* and *c* are the parameters to be estimated.

The coefficient of adjusted determination ($R^2$) is the most widely used criterion in the biomass model literature. The mean absolute prediction error (*MAPE*) was applied as the primary metric to evaluate the performance of models, whose statistical characteristics are proverbial and frequently used in ecology and environment assessment. The selection of our final model was based on high adjusted $R^2$ and low *MAPE*. The $R^2$ and *MAPE* were computed as follows:

$$R^2 = 1 - \frac{\sum_{i=1}^{n}(M_i - \hat{M}_i)^2}{\sum_{i=1}^{n}(M_i - M_i)^2}$$

$$MAPE = \frac{1}{n}\sum_{i=1}^{n}\frac{|M_i - \hat{M}_i|}{M_i}$$

where $M_i$ is observed biomass, $\hat{M}_i$ is predicted biomass, $M$ is the mean of observed biomass, and $n$ is the number of trees.

In this study, we first used the above Equations (4)–(6) to estimate the biomass of tree components, including leaf, branch and stem. Then, we selected the best equation according to the evaluation statistics ($R^2$ and *MAPE*). Next, a linear seemingly unrelated regression approach (SUR) was used to fit the system of selected model that ensured the additivity property. The additivity of the linear equations is enforced by setting a constraint on the regression coefficients. The primary result showed that Equation (5) can better improve the fitting effect and performance of the model. Therefore, the following additive model system was constructed through a linear SUR based on the log-transformed data:

$$\ln(M_l) = a_l + b_l \times \ln(DBH) + c_l \times \ln(H) + \varepsilon_l \tag{7}$$

$$\ln(M_s) = a_s + b_s \times \ln(DBH) + c_s \times \ln(H) + \varepsilon_s \tag{8}$$

$$\ln(M_b) = a_b + b_b \times \ln(DBH) + c_b \times \ln(H) + \varepsilon_b \tag{9}$$

$$\ln(M_{AGB}) = \ln(M_l) + \ln(M_s) + \ln(M_b) + \varepsilon_{AGB} \tag{10}$$

where the subscripts l, s, b and AGB stand for leaf, stem, branch and aboveground, respectively. $\varepsilon$ is the model correction factor.

Root biomass in different *T. Zhongshanshan* stands was estimated using a fixed root to shoot ratio (0.26) as described by Ravindranath and Ostwald [15]. Biomass in all of the ecosystem components was extrapolated and scaled to a per hectare basis.

### 2.6. Estimation of Ecosystem Carbon Storage

We determined the carbon storages in plants by multiplying carbon concentration with dry mass amount. SOC storage up to 60 cm depth was calculated using SOC concentration, bulk density, and soil depth as follows:

$$\text{SOC storage (t ha}^{-1}) = \text{SOC} \times \text{BD} \times \text{T} \times 100$$

where SOC is soil organic carbon (%), BD is bulk density (Mg m$^{-3}$) and T is the soil thickness (m). The ecosystem carbon storage was calculated by adding biomass carbon storage (*T. Zhongshanshan* plants, herbs, shrubs, and litter) and SOC storage.

### 2.7. Statistical Analyses

The log-transformed linear regression procedure in SPSS 19.0 (IBM Corporation, Somers, NY, USA) software was used to fit the *H-DBH* function and allometric models' parameters. The procedure fits model parameters and variance parameters simultaneously by applying the maximum likelihood regression approach. This category of procedure was used due to its flexibility to work with equations forms and its recognized robustness over nonlinear models with additive error and log-transformed models. The SUR in the SAS/ETS Model Procedure (SAS Institute, Inc., Cary, NC, USA, 2011) [16] was used to fit

the system of biomass equations for *T*. Zhongshanshan, in which the coefficients of the tree component biomass models were simultaneously estimated. The plotting software was Origin 2021 (Origin Lab, Northampton, MA, USA).

## 3. Results

### 3.1. Height-Diameter at Breast Height Function

The *H-DBH* function derived from the harvest trees and 5 stands along the Yangtze River were presented in Figure 2, with a correlation coefficient (*r*) of 0.934. Maximum tree heights occurred at *DBH*s of 35–45 cm, and all were of 22-year-old *T*. Zhongshanshan plantation. The predicted *H-DBH* function curve revealed that tree *H* generally showed a positive correlation with the increase in *DBH*, regardless of the site. This *H-DBH* function overestimated trees larger than *DBH* of about 35 cm for approximately 0.2 m in *H*.

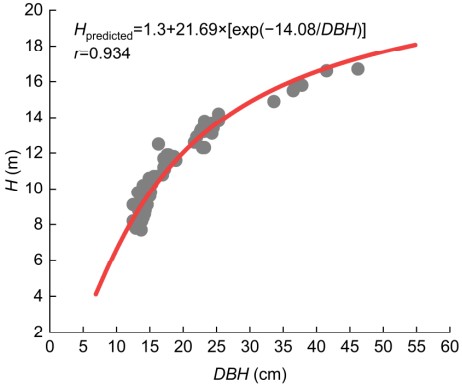

**Figure 2.** Observed scatter and predicted *H-DBH* function curve of *T*. Zhongshanshan plants.

### 3.2. Each Component Biomass Model Selection

The allometric equation in the form of a polynomial function is applicable to calculate the biomass of individual tree components. For *T*. Zhongshanshan stands, Equation (5) performed the best in estimating the leaf, stem and branch biomass compared to the other two models in terms of the $R^2$ and *MAPE* (Table 2). Thus, we used the Equation (5) to model the leaf, stem and branch biomass. The second performance model is Equation (6), with $R^2$ and *MAPE* slightly lower and higher than Equation (5), respectively.

**Table 2.** Equation parameters (*a*, *b* and *c*) and evaluation statistics ($R^2$ and *MAPE*) of allometric models for predicting individual components of *T*. Zhongshanshan plants.

| Equation | Model | Component | Estimated Coefficients | | | $R^2$ | MAPE |
|---|---|---|---|---|---|---|---|
| | | | *a* | *b* | *c* | | |
| (4) | $\ln(M) = a + b \times \ln(DBH)$ | leaf | −2.08 | 1.52 | - | 0.847 | 0.0999 |
| | | stem | −1.88 | 2.15 | - | 0.972 | 0.0255 |
| | | branch | −2.95 | 2.22 | - | 0.971 | 0.0278 |
| (5) | $\ln(M) = a + b \times \ln(DBH) + c \times \ln(H)$ | leaf | −3.57 | 0.16 | 2.47 | 0.853 | 0.0970 |
| | | stem | −4.02 | 0.18 | 3.57 | 0.979 | 0.0225 |
| | | branch | −5.63 | 0.01 | 4.01 | 0.980 | 0.0276 |
| (6) | $\ln M = a + b \times \ln[(DBH)^2 \times H]$ | leaf | −2.45 | 0.59 | - | 0.850 | 0.0992 |
| | | stem | −2.40 | 0.84 | - | 0.975 | 0.0247 |
| | | branch | −3.49 | 0.87 | - | 0.975 | 0.0275 |

### 3.3. Additive Biomass Equations

For the SUR approach, the model for biomass components was fitted to enforce the additivity of the total tree biomass, including the leaf, stem and branch (Table 3). The SUR approach consisted first of fitting and selecting the best model (Equation (5) in this study) for each tree component. In *T*. Zhongshanshan plantations, the fit accuracy of the

leaf and stem biomass was higher than Equation (5) by higher adjusted $R^2$ and lower *MAPE*. However, the branch biomass prediction using SUR method performed worse than Equation (5), which was accomplished by the maximum likelihood regression approach. Based on the SUR model for each component analyzed, we used $M_t$ (total biomass) = $M_l$ + $M_s$ + $M_b$ + $M_r$ (root biomass) to calculate each component tree biomass. For leaf, stem and branch, the biomass values predicted by the maximum likelihood regression were close to the observed values, and the generalized SUR approach had similar estimations of leaf and stem but superior evaluation statistics (Figure 3).

**Table 3.** Parameter estimates (*a*, *b* and *c*) and evaluation statistics ($R^2$ and *MAPE*) of Equation (5) for different components using SUR approach.

| Component | Estimated Coefficients | | | $R^2$ | *MAPE* |
|---|---|---|---|---|---|
| | *a* | *b* | *c* | | |
| leaf | −3.66 | 0.69 | 2.15 | 0.892 | 0.0927 |
| stem | −3.92 | 0.18 | 3.57 | 0.981 | 0.0197 |
| branch | −4.62 | 0.05 | 3.59 | 0.953 | 0.0495 |

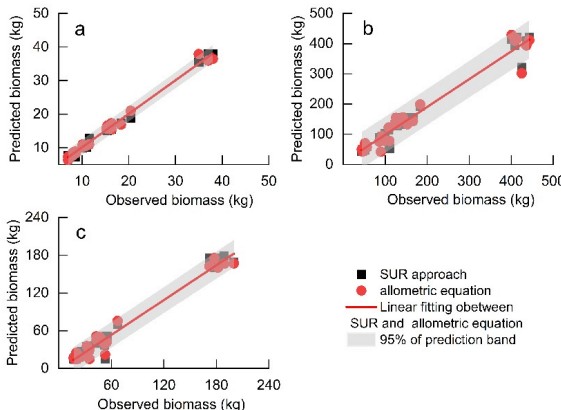

**Figure 3.** Comparison of the predicted leaf (**a**), stem (**b**) and branch (**c**) biomass against the corresponding observed biomass in *T*. Zhongshanshan plantations by SUR approach and allometric equation, respectively.

### 3.4. Individual Tree Biomass and Allocation

Adopting the equations established by SUR, the biomass of individual tree components was determined in the 9-, 11-, 13-, 15-, and 22-year-old *T*. Zhongshanshan plants (Figure 4). With increasing stand age, the biomass of all tree components increased. The total individual tree, leaf, stem, branch and root biomass was 156.80, 15.27, 64.20, 23.64 and 26.82 kg of the 9-year-old *T*. Zhongshanshan plants and increased to1245.72, 93.5, 545.50, 181.01 and 213.08 kg of the 22-year-old plants. With increasing stand age, the proportion of leaf biomass gradually decreased with values of 9.74%, 8.40% and 7.36% in the 9-, 11- and 13-year-old *T*. Zhongshanshan plants. For all *T*. Zhongshanshan plants, the stem had the largest proportion of biomass among all components, whose contribution to total tree biomass was 40.94%, 41.95%, 43.00%, 43.38% and 43.79% along the chronosequence, showing a slightly increasing trend. Similar to the stem, individual branch biomass also exhibited a slightly increasing tendency with increasing stand age.

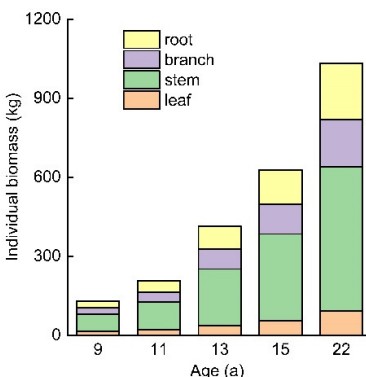

**Figure 4.** Individual biomass of each component of the 9-, 11-, 13-, 15-, and 22-year-old *T.* Zhongshanshan plants.

According to the plant density, the biomass of different ecosystem components was established in Table 4. As the stand aged, the biomass of most tree components increased, including total tree, leaf, stem, branch and root biomass, except for the 22-year-old *T.* Zhongshanshan stand. Since the density of the above-mentioned stand was 410 stems·ha$^{-1}$, which was less than half of the other stands. The total tree biomass was 53.43, 84.87, 140.67, 192.71 and 156.65 t ha$^{-1}$ in the 9-, 11-, 13-, 15-, and 22-year-old *T.* Zhongshanshan stands. Total understory biomass exhibited a slight increment from 9 to 15-year-old stand and was 3.68 times compared to 15-year-old stand. Litter biomass decreased from 9-year-old stand (3.26 t ha$^{-1}$) to 13-year-old stand (3.12 t ha$^{-1}$) and increased in later stages (6.21 and 6.12 t ha$^{-1}$, respectively, in 15- and 22- year-old stands).

**Table 4.** Biomass of different ecosystem components in the 9-, 11-, 13-, 15-, and 22-year-old *T.* Zhongshanshan stands.

| Components | Biomass (t ha$^{-1}$) | | | | |
|---|---|---|---|---|---|
| | **9-Year-Old** | **11-Year-Old** | **13-Year-Old** | **15-Year-Old** | **22-Year-Old** |
| Leaf | 6.28 | 8.60 | 12.50 | 17.12 | 14.12 |
| Stem | 26.39 | 42.95 | 72.99 | 100.86 | 82.75 |
| Branch | 9.74 | 15.81 | 26.19 | 34.97 | 27.46 |
| Root | 11.03 | 17.51 | 29.00 | 39.77 | 32.32 |
| Total tree | 53.43 | 84.87 | 140.67 | 192.71 | 156.65 |
| Total understories | 1.82 | 1.99 | 2.21 | 2.22 | 8.15 |
| Litter | 3.27 | 2.14 | 3.12 | 6.22 | 6.12 |
| Total | 58.52 | 88.99 | 146.01 | 201.15 | 170.92 |

The total ecosystem biomass of *T.* Zhongshanshan stands over the 5 age groups was highest in the stem, followed by (in decreasing order) the root, the branch, the leaf, understories or litter. The proportion of the biomass from stem increased with forest age (except for the 22-year-old *T.* Zhongshanshan stand), while that in the leaf and understories declined. The stem accounted for 48.87%, 50.80%, 51.95% and 58.93% in the 9-, 11-, 13-, 15-year-old stand, respectively. The leaf accounted for 10.73%, 9.67%, 8.56% and 8.51%, the understory for 3.12%, 2.23%, 1.51% and 1.10% from 9 to 15-year-old stand, respectively. Although the root biomass of *T.* Zhongshanshan was derived from fixed root to shoot ratio (0.26), the highest proportion occurred in 15-year-old with 19.99%, which was seemingly independent of tree age in this study. Since root biomass proportion of total ecosystem was 18.84%, 19.68%, 19.86%, 19.77% and 18.91% from 9 to 22-year-old stand, respectively (Figure 5).

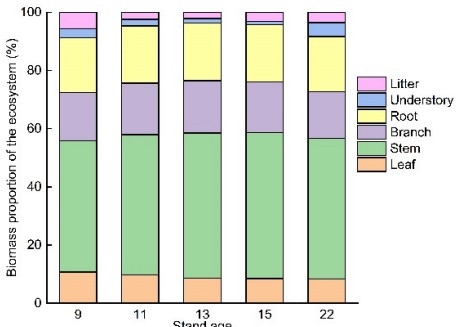

**Figure 5.** Biomass proportion of the ecosystem from leaf, stem, branch, root, understory and litter in the 9-, 11-, 13-, 15-, and 22-year-old *T*. Zhongshanshan stands.

### 3.5. Ecosystem Carbon Storage and Allocation

The aboveground (leaf, stem, branch, understory and litter), belowground (root), and total ecosystem carbon storage for each of the 5 stands were provided in Table 5. The carbon storage of the total tree, soil, and total ecosystem increased with increasing stand age (except for the 22-year-old *T*. Zhongshanshan stand), from 25.32, 96.89 and 124.76 t ha$^{-1}$ in the 9-year-old stand, respectively, to 90.89, 117.12, and 212.22 t ha$^{-1}$, respectively, in the 15-year-old stand. In the 22-year-old *T*. Zhongshanshan stand, carbon storage was 73.85, 136.66 and 217.64 t ha$^{-1}$, respectively, from total tree, soil and total ecosystem. For total tree carbon storage manifested a sigmoidal schema, first promptly rising and then gradually decreasing. The carbon sequestration rates of the ecosystem were 10.26 and 12.26 t ha$^{-1}$ year$^{-1}$ during the 9th–13th and 13th–15th years, respectively. Compared with the 9-year-old stand, the carbon storage of the 0–60 cm soil layer increased from 11 to 22-year-old stand, indicating an evident accumulation course of organic carbon after afforestation, with an annual accumulation rate of 3.05 t ha$^{-1}$. The understory carbon storage (4.07 t ha$^{-1}$) of the 22-year-old stand was much higher than that of the other stands (0.91–1.10 t ha$^{-1}$), indicating an obvious increase in floor vegetation. The carbon storage of the litter exhibited a distinct relationship with stand age, with lower values (1.06–1.63 t ha$^{-1}$) and higher values (3.06–3.10 t ha$^{-1}$) occurring in the 9 to 13-year-old stands and 15 to 22-year-old stands, respectively.

**Table 5.** Carbon storage of different ecosystem components in the 9-, 11-, 13-, 15-, and 22-year-old *T*. Zhongshanshan stands.

| Components | Carbon Storage (t ha$^{-1}$) | | | | |
|---|---|---|---|---|---|
| | **9-Year-Old** | **11-Year-Old** | **13-Year-Old** | **15-Year-Old** | **22-Year-Old** |
| Leaf | 3.26 | 4.47 | 6.50 | 8.90 | 7.34 |
| Stem | 11.59 | 18.85 | 32.04 | 44.28 | 36.33 |
| Branch | 4.74 | 7.70 | 12.75 | 17.03 | 13.37 |
| Root | 5.73 | 9.11 | 15.08 | 20.68 | 16.81 |
| Total tree | 25.32 | 40.13 | 66.37 | 90.89 | 73.85 |
| Understory | 0.91 | 0.99 | 1.11 | 1.11 | 4.07 |
| Litter | 1.63 | 1.07 | 1.56 | 3.11 | 3.06 |
| Soil | 96.89 | 104.36 | 116.32 | 117.12 | 136.66 |
| Total ecosystem | 120.08 | 156.68 | 211.74 | 263.11 | 231.49 |

In all 5 *T*. Zhongshanshan stands, the proportion of carbon storage in the soil was more than half, with specific values of 77.66%, 71.21%, 62.75%, 55.19%, and 62.79%, respectively (Figure 6). Except for in the 22-year-old stand, the proportion of carbon stored in the soil decreased with increasing age and the proportion stored in vegetation increased correspondingly. Of the vegetation parts, stem and root were the two largest contributors to the total ecosystem carbon pool in all 5 stands. Understory and litter contributed little to

the total site carbon storage, accounting from 0.52% to 1.87%, and 0.73% to 1.46% within these stands. The contribution of the tree root increased with stand aging from 4.60% in the 9-year-old stand to 8.14% in the 13-year-old stand, and then decreased to 9.74% and 7.72% in the 15- and 22-year-old stands.

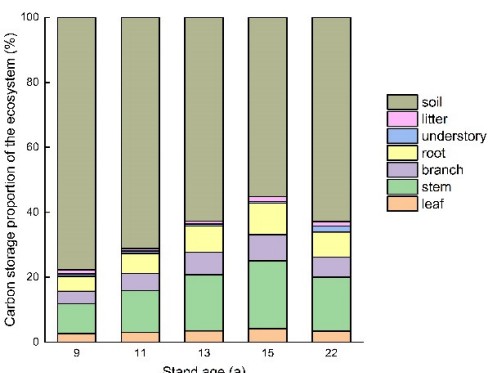

**Figure 6.** Carbon storage proportion of the ecosystem from leaf, stem, branch, root, understory and litter in the 9-, 11-, 13-, 15-, and 22-year-old *T*. Zhongshanshan stands.

## 4. Discussion

### 4.1. Tree Biomass Development

China, especially the YRB area, has a stringent policy of forest protection [17,18] and it was, hence, impossible to log trees for our study to establish site-specific allometric models. Moreover, the number of logged *T*. Zhongshanshan trees was low, owing to the conservation of germplasm resources limitation and scarcity of *T*. Zhongshanshan older than 20 years. In this study, only a limited number of samples of *T*. Zhongshanshan material was available for the construction of the biomass equations across the different stages of stand development. The quantity of tree samples used to develop allometric equations is quite multitudinous in the literature. Xue et al. [19] harvested 72 *Casuarina equisetifolia* trees on Hainan Island; Mugasha et al. [20] reported 30 samples for the wet lowland forests equations; Zhang et al. [21] established allometric biomass equations based on 8 Mongolian Pine trees. The precision of allometric models is usually determined by pivotal factors, such as tree species, *DBH* homogeneity, or the amount of sampled trees. For *T*. Zhongshanshan stands, artificially planted forests have consistent age and growth performance, which can make up for the shortage of quantity to a certain extent.

Previous studies have demonstrated that a single *DBH*-based allometric model can reasonably predict biomass in several plants including *Pentaclethram acroloba* [22], *Pinus tabuliformis* [23] and *Cunninghamia lanceolata* [24]. As before, ordinary surveys of diameter were not only simpler to implement in the field but were also more often to exist in historical data. For *T*. Zhongshanshan plants, although *DBH* evidenced to be a fairly good predictor of biomass, the selected 3-parameter polynomial model performed better. On this basis, the implementation of SUR approach provided further accurate estimates of biomass component fractions. Such results are consistent with many previous studies, which showed aboveground biomass models with combined *DBH* and *H* as the most suitable predictors [25–27]. Chave et al. [25] and Mensah et al. [26] both verified that the inclusion of *H* reduced the error of aboveground biomass estimates by 6.70% and 35.30% in predicting the biomass of forests, since *H-DBH* relevance varied across a range of ecological conditions. In West African savanna ecosystems, Ganamé et al. [28] narrowed the confidence intervals of the biomass estimation and subsequently increases the accuracy of the predictions by applying SUR approach. Diverse from Mbow et al. [29], who refrained from relying heavily on tree *H* and hold the opinion that the accuracy of tree measurement was generally much lower than that of *DBH*. This is often due to the approximate methods adopted to estimate *H* in the early field investigation, simultaneously the rather arbitrary condition to consider, such as diminutive and isolated branches stretching out of the canopy.

With the improvement of *H* measurement accuracy, the estimation of tree biomass, as well as the description of stands and their performance over time, relies largely on an accurate *H-DBH* relationship [30]. We also developed *H-DBH* function derived from the harvest trees and 5 stands along the Yangtze River. This is particularly beneficial to *T.* Zhongshanshan plantations in the cases where the actual measurements of *H* growth are not available, and follow-up estimating biomass across large spatial scales using forest inventory data.

*4.2. Biomass Distribution*

Age is the most vital factor influencing the magnitude and distribution of biomass in plants. In this study, the total individual tree, leaf, stem, branch and root biomass increased promptly from the 9th to the 22nd year, where a linear positive correlation could be extracted for this species between forest age and total tree biomass. The proportion of leaf biomass gradually decreased with increased stand age. This result was consistent with the reports of many previous studies [31,32]. Acting as a valuable component, the leaf is highly correlated with forest performance in young *T.* Zhongshanshan stands (<10-year-old). As stands age, tree magnitude increases during ontogeny and more carbohydrate resources are distributed for stem growth [19]. Hence, leaf biomass increases proportionally less than stem mass. The proportion of stem accounted for most, and remained stable at later stages, indicating that stem is a vital composition when accounting biomass partitioning for *T.* Zhongshanshan plants [33].

Extrapolated by the polynomial growth equation, the total tree biomass of *T.* Zhongshanshan increased rapidly in the first 2 or 3 decades and increased slowly later, showing a sigmoidal pattern. Such a pattern after afforestation could also be found in many prior studies [23,24,29]. From the 9- to the 15-year-old stand in this study, the *T.* Zhongshanshan ecosystem biomass increased steadily, and there was a slight decline in the 22-year-old stand, which could be owing to a smaller forestation density (approximately 5.4 m and 4.5 m spacing between and within rows). In *P. strobus* plantations, the biomass in the 42-year-old stand corresponds to that of the 65-year-old, implying that tree density played an important role in ecosystem biomass accumulation [34]. Thereupon, understory biomass in the 22-year-old stand was nearly 4 times that of the 9- to 15-year-old stands averagely. As demonstrated by Xu et al. [35], the coverage and biomass of the understory vegetation increased significantly with the growth of the stand. These consistent relationships between understory biomass and age might be due to the significant increase in canopy closure, which may result in species diversity and abundance. As forest aged, a smaller initial tree density might increase the light transmittance for understory plants and lead to lesser competition. A more spacious interval coupled with light penetration help to increase temperature and ventilation, which is instrumental to understory growth [36]. The total tree biomass was 192.71 and 156.65 t ha$^{-1}$ in the 15- and 22-year-old *T.* Zhongshanshan stands. In comparison with 15-year-old *P. massoniana* [37] and 22-year-old *P. strobus* stands [34], whose tree biomass was 78.5 and 128.0 t ha$^{-1}$, *T.* Zhongshanshan stands of the same age exhibited higher biomass accumulation. Although Moriondo et al. [38] reported that tree biomass accumulation depends on growth habitat, the soil on which plants are growing, disturbance regime and interaction with belowground vegetation, our study proved that *T.* Zhongshanshan is a fast-growing variety to some extent.

*4.3. Ecosystem Carbon Storage*

Ecosystem carbon storage and portioning accorded well with biomass accumulation in the vegetation component and were age-dependent in the soil component. The total ecosystem carbon storage of 9- to 22-year-old *T.* Zhongshanshan stands ranged from 124.76 to 217.64 t ha$^{-1}$, of which total tree ranges from 25.32 to 90.89 t ha$^{-1}$. These values were well within the range of above-ground biomass carbon storage (4.5–462.5 t ha$^{-1}$, averagely 61.9 t ha$^{-1}$) reported by a case study of carbon sequestration following reforestation in the YRB area [39]. *T.* Zhongshanshan plantations had a substantial potential for carbon sequestration as the total land area under these plantations was consecutively expanding,

most of which was still immature. The carbon storage of the 0–60 cm soil layer increased from 11- to 22-year-old stand, with an annual accumulation rate of 3.05 t ha$^{-1}$, implying substantial amounts of carbon accumulation during the transition period from young to near-mature (20- to 30-year-old) *T*. Zhongshanshan stands. The older age of tree stands produces more litter and root biomass which ultimately supplies more organic matter to the soil. Our results of SOC storage were comparable to the *T. distichum* forests (123.3 t ha$^{-1}$) of South Caspian Sea in similar soil depth [40] and were higher than the national average in China (96.0 t ha$^{-1}$) [41] but lower than upper reaches of the YRB (164 t ha$^{-1}$) [42]. Probably owing to the regional climatic patterns, a decreasing temperature from west to east in the YRB led to a slower decomposition of SOM and a lower soil respiration rate in the upper reaches. Corresponding to other investigations conducted in forest ecosystems [43], the highest carbon storage occurred in the soil component of *T*. Zhongshanshan ecosystem but contradicted the findings of Vesterdal et al. [44] and Lü et al. [45], who found that soils only contributed about one third in an afforestation ecosystem. Gogoi et al. [46] proclaimed that the continuous disturbance by human interference declines SOC storage. The starting point in soil carbon storage also possibly determined the sequestration potential.

## 5. Conclusions

The results described by the selected 3-parameter polynomial model and SUR approach pave the way for a more systematic estimation of *T*. Zhongshanshan biomass. The fitted *H-DBH* function is quite capable of accounting for the relationship between *H* and *DBH* of *T*. Zhongshanshan plants. *T*. Zhongshanshan is a fast-growing variety, as compared tree biomass with other trees of the same stand age in similar subtropical areas. In conclusion, the present study revealed that the *T*. Zhongshanshan stands in the YRB area can store 124.76 to 217.64 t ha$^{-1}$ carbon, of which total tree ranges from 25.32 to 90.89 t ha$^{-1}$, with 55.19% to 77.66% storing in the soil. Large *T*. Zhongshanshan trees contributed greatly to carbon storage in living biomass and may be the main reason accounting for having a persistent potency. A potential limitation involved with this study was the absence of tree age and taper as explanatory variables. The performance of the generalized approaches in subsequent studies could be marginally improved on a large enough and sufficiently representative set of individual plants by combining more explanatory variables. Multisite studies are further required to fully elaborate the patterns of biomass and carbon storage of individual trees and the ecosystem of *T*. Zhongshanshan plantations by stand age.

**Author Contributions:** Q.S.: conceptualization, data collection, formal analysis, writing—original draft, editing; J.H.: conceptualization, writing—review and editing; D.C.: writing—review and editing; Y.Y.: writing—review and editing. All authors have read and agreed to the published version of the manuscript.

**Funding:** This research was funded by the Innovation Project of Plant Germplasm Resources of Chinese Academy of Sciences (kfj-brsn-2018-6-003), Jiangsu Special Fund on Technology Innovation of Carbon Dioxide Peaking and Carbon Neutrality (BE2022420), and Jiangsu Long-term Scientific Research Base for Taxodium Rich. Breeding and Cultivation (LYKJ [2021]05).

**Data Availability Statement:** Publicly available datasets were analyzed in this study.

**Conflicts of Interest:** The authors declare no conflict of interest.

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
