# Peer review of "Biomass Estimation and Carbon Storage of Taxodium Hybrid Zhongshanshan Plantations in the Yangtze River Basin"

_forests, doi:10.3390/f13101725_

Round 1
Reviewer 1 Report
Introduction:
46-47 lines
This claim is controversial. Most biomass estimation research is based on compatible systems. These systems use the biomass of the fractions to estimate the total biomass without bias. (https://www.scielo.cl/scielo.php?pid=S0717-92002021000100053&script=sci_arttext).
It is not clear if you will use a system or not.
Materials and Methods
Line 174: There you explain that you use a system of equation.
Perhaps you can review the previous lines of the introduction and explain the advantages of these compatible systems.
Results
Figure 2. You should improve the graph and in the model replace H with estimated H.
Table 2. The caption is incomplete. It does not refer to the quality statistics of the fit .
Lines 214-215. You do not repeat the information (table 3). Replaces 980 by 0.980
Figure. 3. Could you improve the graph and explain its interpretation.
Point 3.3. The use of the comptible system remains unclear. If you don't use a fit method like SUR, FIML, 2SLS, etc., your system will not have consistent and compatible parameters.
Figure 5. I recommend that you organize and draw the graphs so that the work remains in the same style.
Line 269. Do you mean table 4?
In general:
There are a lot of biomass estimation research in different species. You should innovate your research.
Based on your results, some observations that could improve and enrich your research.
Did you check the relevance of a biomass estimation model with age as an explanatory variable? or the use of a mixed model according to age/fbiomass fraction?
Did the differences in biomass at each age respond to different volumes or different shapes of the trees? Did you try to fit a taper equatio?
How was the relationship between taper, volume and biomass? Will age improve taper systems?
Reviewer 2 Report
I have reviewed the manuscript title “Biomass estimation and carbon storage of Taxodium hybrid Zhongshanshan plantations in the Yangtze River Basin” thoroughly and it has worth accepting after the revisions proposed below.
Why you have selected Taxodium hybrid Zhongshanshan?
What is the botanical name of this plant?
Why you have taken 25 X 25 m plot? Please explain.
How you have taken 25 X 25 m plot? Please explain.
What is the purpose “Tree biomass model development”? Please explain. Why you didn’t use the already available models?
Maximum tree heights occurred at DBHs of 35-45 cm, and all were of 22-year-old T. Zhongshanshan plantation. What is the reason?
For T. Zhongshanshan stands, equation 5 performed the best in estimating the leaf, stem and branch biomass compared to the other two models in terms of the R2 and MAPE. What is the reason?
With increasing stand age, the proportion of leaf biomass decreased. What is the possible reason?
As the stand aged, the biomass of most tree components increased, including total tree, leaf, stem, branch and root biomass, except for the 22-year-old T. Zhongshanshan stand. What is the possible reason?
The total ecosystem biomass of T. Zhongshanshan stands over the 5 age groups was highest in the stem, followed by (in decreasing order) root or branch, leaf, understories or litter. What is the reason?
For total tree and total ecosystem, carbon storage manifested a sigmoidal schema, first promptly rising and then gradually decreasing. What is the reason?
Except for in the 22-year-old stand, the proportion of carbon stored in soil decreased with increasing age and the proportion stored in vegetation increased correspondingly. What is the reason?
In T. Zhongshanshan plants, although the main predictor of biomass, DBH, tended to perform well for predicting biomass, the selected 3-parameter polynomial model (M=a×(DBH)b×Hc) performed better, and it provided accurate estimates of all biomass component fractions. Rewrite this sentence.
Line number 327-328: Such results are consistent with many previous studies, which showed aboveground biomass models with combined DBH and H as the most suitable predictors. Which studies? Please cite the studies.
Line Number 347-348: With increasing stand age, the proportion of leaf biomass decreased gradually in this study, which was consistent with the reports of many previous trees. Please re-write this sentence.
Line Number 349-353: Acting as a valuable component, the leaf is highly correlated with forest performance in young T. Zhongshanshan stands (<10-year-old), which typically peak as canopies close and then decrease with stand age. The proportion of stem accounted for most, and remained stable at later stages, indicating that stem is a vital composition when accounting biomass partitioning for T. Zhongshanshan plants. Please re-write this sentence. Also, provide evidence from the literature.
Line Number 358-360: From the 9- to the 15-year-old stand in this study, the T. Zhongshanshan ecosystem biomass increased steadily, and there was a slight decline in the 22-year-old stand, which could be owing to smaller forestation density (approximately 5.4 m and 4.5 m spacing between and within rows). On what basis you are saying this?
Round 2
Reviewer 1 Report
I agree with the modifications and the changes made